# Distribution of Zooplankton Functional Groups in the Chaohu Lake Basin, China

Li Wu [1,2], Lei Ji [3,*], Xiaojuan Chen [1,*], Jiajia Ni [4], Yan Zhang [2] and Ming Geng [2]

1 Key Laboratory of Ecological Impacts of Hydraulic-Projects and Restoration of Aquatic Ecosystem of Ministry of Water Resources, Institute of Hydroecology, Ministry of Water Resources and Chinese Academy of Sciences, Wuhan 430079, China; wuli090121@126.com

2 School of Life Sciences, Hefei Normal University, Hefei 230061, China; zhangyanwind@163.com (Y.Z.); gm7093@sina.com (M.G.)

3 Anhui Province Key Laboratory of Pollutant Sensitive Materials and Environmental Remediation, College of Life Science, Huaibei Normal University, Huaibei 235000, China

4 Research and Development Center, Guangdong Meilikang Bio-Science Ltd., Dongguan 523808, China; nijiajia2005@126.com

* Correspondence: 626jl@163.com (L.J.); chenxiaojuan@mail.ihe.ac.cn (X.C.)

**Abstract:** To analyze the structural characteristics of zooplankton functional groups (ZFGs) and their correlation with environmental physicochemical factors in the Chaohu Lake Basin, water samples were collected from October 2019 to July 2020, and the zooplankton species and ZFGs were investigated. A total of 250 species, including 88 protozoa, 115 rotifers, 28 cladocerans, and 19 copepod species, were detected and divided into 16 ZFGs. The ZFGs exhibited obvious spatiotemporal heterogeneity. ZFGs in the Chaohu Lake were notably different from those in rivers and were different between the rivers. In the ecosystem, network analysis showed that protozoan algae/protozoan bacteria (PA/PB), rotifer particle filter (RF), and rotifer small predator (RSG) were important in the spring, summer, and autumn that and small zooplankton filter (SCF) was important in spring, autumn, and winter, while the importance of other ZFGs changed with seasons. Redundancy analysis showed that the environmental factors with a strong correlation between the ZFG compositions differed in each season. Different ZFGs exhibited different correlations with environmental factors. This study showed that ZFGs were closely related to environmental factors and that functional traits can reflect responses to changes in the water environment.

**Keywords:** Chaohu Lake Basin; zooplankton; functional group; environmental factors; freshwater ecology





## 1. Introduction

Zooplanktons are an important component of freshwater ecosystems serving important ecological functions. They are the link for energy and nutrient flow between small producers, such as phytoplankton, and large secondary consumers, such as fish [1,2]. Zooplanktons are important biological indicators for environmental monitoring as they respond quickly to natural or artificial environmental changes [3–5]. However, a zooplankton-diversity-index-based evaluation of water environment can only reflect a subset of the environmental state, and the interpretation of this conclusion is subjective [6].

Functional characteristics are the interactions between organisms and ecosystems [7], and their analysis can improve the assessment of their responses to environmental changes [8]. Functional group (FG) refers to a set of similar response species in a specific environment or habitat [9]. FGs reflect ecological processes more directly than zooplankton species do. As a result, dividing communities into FGs is an ideal method for conducting aquatic ecological research. The responses of FGs to environmental changes are more comprehensive than those of individuals and populations [10]. FG analyses have been widely used in the study

of phytoplankton communities [11,12], which can aid in a better understanding of the functional changes in ecosystems. However, research on zooplankton FGs (ZFGs) is currently restricted to a few groups [13–16], with only a few studies focusing on ZFGs in freshwater ecosystems. [17,18].

The Chaohu Lake, one of Central China's five largest freshwater lakes, is a typical shallow lake downstream of the Yangtze River. There are 33 rivers in the Chaohu Lake Basin that belong to seven water systems, including the Zhegao River (ZGR), Nanfei-Dianbu River (NFR), Pai River (PR), Hangbu-Fengle River (HBR), Baishishan River (BSSR), Zhao River (ZR), and Yuxi River (YXR), with the YXR flowing out of the lake and connecting to the Yangtze River and the other six rivers flowing into the lake (Figure 1) [19]. The landscape of the Chaohu Lake Baisn is primarily plains, and the elevation decreases from west to east, ranging from 8 to $1.49 \times 10^3$ m in low mountains. Land use is dominated by farmland (about 70%) in the catchment, followed by urban land to the west and east of the lakeshore [20]. Therefore, NFR is urban industrial catchments, HBR upstream is a natural catchment, and other river and lake areas are agricultural catchments. With the rapid industrialization and urbanization in the Chaohu Lake Basin in recent years, the water of the Chaohu Lake has suffered severe eutrophication. A number of serious ecological and environmental issues, such as deterioration of water quality and decline of lake ecosystems, have arisen due to outbreaks of cyanobacteria in lakes. For instance, external loading of Lake Chaohu, especially by the Nanfei and Hangbu Rivers, contributed remarkably to cyanobacterial blooms [21]. However, most zooplankton research in the Chaohu Lake Basin is currently focused on community structure and water quality [22,23], with little attention paid to ZFGs. To analyze the characteristics of ZFGs, their spatiotemporal succession, and their correlation with physicochemical factors in the Chaohu Lake Basin, as well as to provide biological data for controlling the Chaohu Lake Basin pollution and water environmental protection, an annual survey of zooplankton in the Chaohu Lake Basin was conducted in this study.

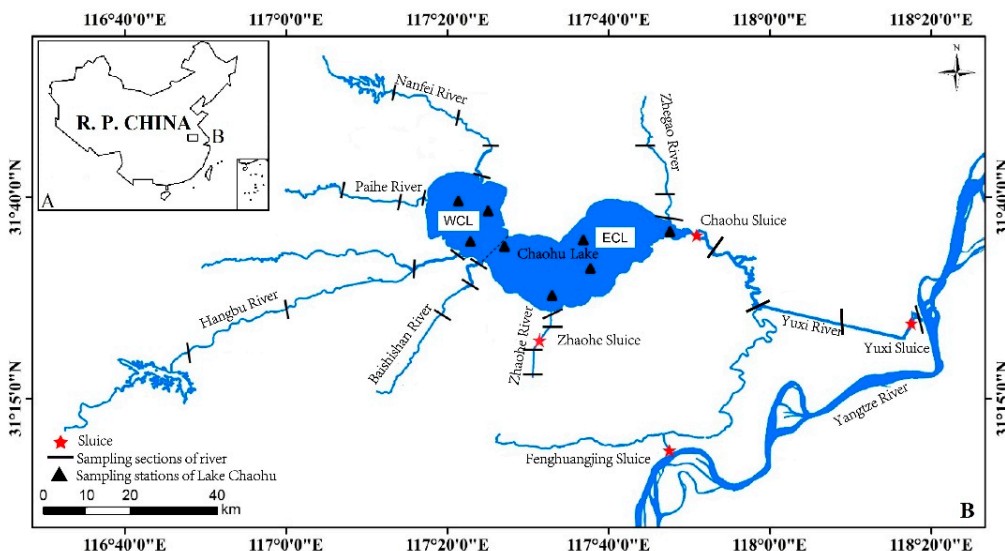

**Figure 1.** Distribution of sampling sites in the Chaohu Lake Basin.

## 2. Materials and Methods

### 2.1. Introduction to the Study Area

The Chaohu Lake Basin (30.8736–32.1314° N, 116.3997–118.3681° E), with $1.35 \times 10^5$ km² of drainage area, is located downstream of the Yangtze River. The area has a subtropical humid monsoon climate with four distinct seasons [20].

### 2.2. Sample Collection

A total of 33 sampling sites were chosen in the Chaohu Lake (eight sampling sites, including four sampling sites each in West Chaohu Lake (WCL) and East Chaohu Lake (ECL) and 25 sampling sites in the seven rivers). The sampling sites were set in the upstream, midstream, downstream, and estuary areas of the NFR, HBR, ZR, and YXR and in the midstream, downstream, and estuary areas of the PR, BSSR, and ZGR. All sampling sites were located using GPS (Figure 1).

Water samples were collected on 14 October 2019 (autumn), 3 January (winter), 24 April (spring), and 10 July 2020 (summer). Based on the water depth, water was collected every 0.5 m of depth using a 1 L water collector, and 1000 mL of mixed water sample was stored. Then, 10 mL Lugol's iodine solution was added to the water sample for quantitative analysis of protozoa and rotifers. Twenty microliters of the mixed water sample was collected using a water collector and filtered using a plankton net (0.064 mm mesh diameter) to obtain 100 mL, and then 2 mL of 40% formaldehyde solution was added for quantitative analysis of cladocerans and copepods. Water temperature (WT), pH, dissolved oxygen (DO), and conductivity (Cond) were measured using a YSI6600-V2 multiparameter water quality analyzer (YSI, Yellow Springs, OH, USA). Transparency (Trans) was measured using a Secchi disk according to the standard method [24]. Approximately 1500 mL of mixed water sample was collected using a water collector at each sampling site, temporarily stored in a storage box with ice bags, and transported to the laboratory within four hours for physicochemical indices.

### 2.3. Species Identification and Counting

Zooplankton species identification was carried out as previously described [6,25–27]. One liter of water from each sampling site was precipitated for 48 h and concentrated to 50 mL for species identification of protozoa and rotifers. Next, 0.1 mL and 1 mL concentrated samples were placed in 0.1 mL (for protozoa count) and 1 mL (for rotifer count) count boxes, respectively; species were then identified and counted under $200\times$ or $400\times$ magnification. Each sample was counted twice, and the average value was calculated. Standard deviation was maintained at <8%. The average value was then converted into the number of individuals per milliliter. Cladocerans and copepods were identified and counted in 20 L mixed water samples collected at each sampling site.

### 2.4. Division of ZFGs

According to previous reports [13,14,17,28], zooplankton in freshwater ecosystems were divided into 20 ZFGs based on their size, feeding habits, and interactions (Table S1). The biomass of each ZFG was calculated by adding the count of the included species.

### 2.5. Determination of Water Physicochemical Properties

Approximately 500 mL water was filtered using a Whatman GF/C filter membrane and used to measure the chlorophyll a content (Chl *a*) as previously described [29]. Total phosphorus (TP), $PO_4$-P, biochemical oxygen demand after five days ($BOD_5$), total nitrogen (TN), permanganate index ($COD_{Mn}$), $NO_3$-N, $NH_4$-N, and $NO_2$-N were determined according to standard methods [30]. Water quality was classified according to $BOD_5$ and $COD_{Mn}$ as per the Environmental Quality Standards for Surface Water (GB 3838-2002).

### 2.6. Data Analysis

One-way ANOVA was implemented using R version 3.3.2. Cluster analysis of the ZFGs was conducted using the PRIMER 6. Redundancy analysis (RDA) was carried out using the vegan package in R to explore the relationship between the density of ZFGs and environmental factors. Co-occurrence network analysis was conducted using the igraph package to analyze the co-occurrences of ZFGs. Statistical significance was set at $p < 0.05$.

## 3. Results

### 3.1. Spatiotemporal Distribution Patterns of Physical and Chemical Indices

Except for $COD_{Mn}$, $BOD_5$, and Chl a, the other environmental factors were significantly different between seasons (one-way ANOVA, $p < 0.05$; Table S2). DO, Cond, TN, $NH_4$-N, $NO_3$-N, and Trans were highest in winter, whereas WT, TP, $PO_4$-P, and $NO_2$-N were highest in summer (Table S2). Except for Chl*a*, the other environmental factors were significantly different among the sampling sites (one-way ANOVA, $p < 0.05$; Table S2). DO and pH were the highest in ECL and WCL. Nutrients (TP, TN, $PO_4$-P, $NO_3$-N, $NO_2$-N, and $NH_4$-N), Cond, $COD_{Mn}$, and $BOD_5$ were highest in the NFR and PR. The trans of the HBR was significantly higher than that of the Chaohu Lake and other rivers ($p < 0.005$; Table S2). Except for the PR water collected in winter and summer, which was class V, the other water types were class III or IV (Figure 2). Clustering analysis showed that the Chaohu Lake and the seven river samples were divided into four groups for each season (Figure 2). The HBR samples were divided into a single group. The NFR and PR samples were clustered into one group in spring, autumn, and winter, but were divided into two completely different groups in summer (Figure 2). The WCL and ECL samples clustered into one group in spring and winter (Figure 2A,D) and clustered with the BSSR, ZR, ZGR, and YXR samples in summer (Figure 2B). The YXR and ZGR samples clustered into one group in autumn (Figure 2C). In total, the Chaohu Lake and the seven rivers exhibited an obvious spatial distribution in the environmental gradient, and the difference in environmental conditions between the HBR and the other samples was the largest, except in summer. WCL and ECL both had similar environmental conditions. The environmental conditions of the NFR and PR were similar. The BSSR, ZHR, ZGR, and YXR exhibited similar environmental conditions. However, the water samples were not clustered completely according to water quality (Figure 2).

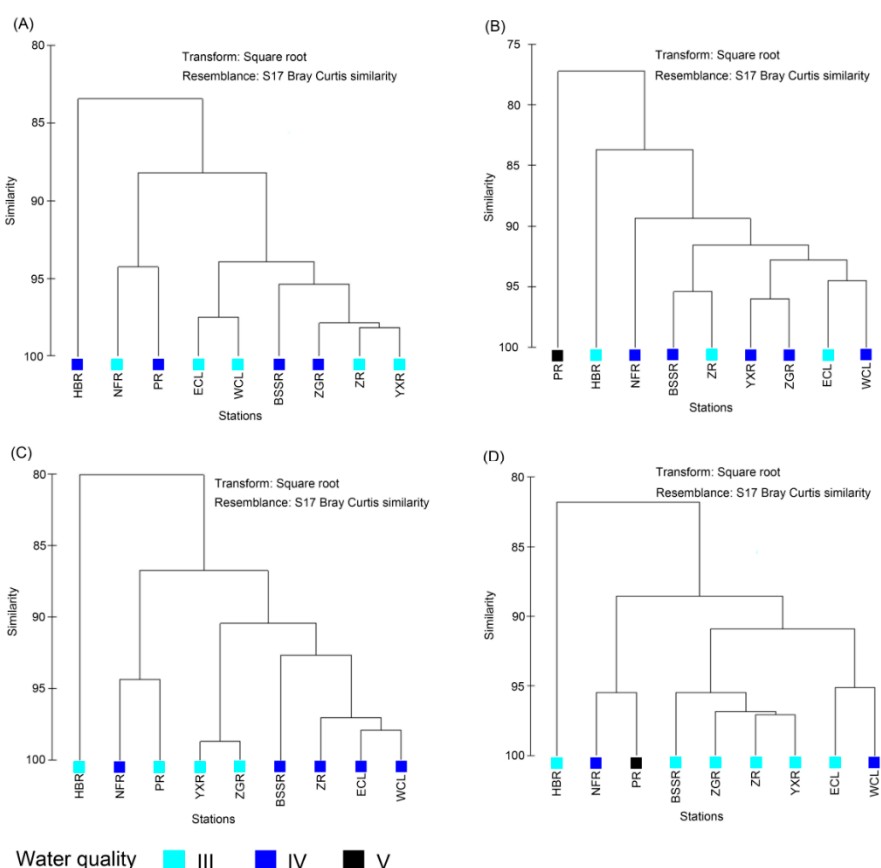

**Figure 2.** Clustering profiles of physicochemical parameters. (**A**) spring, (**B**) summer, (**C**) autumn, and (**D**) winter.

### 3.2. Spatiotemporal Distribution Patterns of ZFGs

A total of 250 zooplankton species were identified, including 88 protozoa, 115 rotifers, 28 cladocerans, and 19 copepods (Table S3). Except for the species that could not be accurately divided into ZFGs, 227 zooplankton species were divided into 16 ZFGs, including protozoan algae (PA), protozoan bacteria (PB), protozoan algae/bacteria-eater (PA/PB), protozoan algae-eater/carnivorous predator (PA/PR), protozoan fungus-eater (PF), protozoan carnivorous predator (PR), rotifer particle filter (RF), rotifer predator (RP), rotifer small predator (RSG), rotifer large predator (RLG), rotifer sucker (RS), small zooplankton filter (SCF), medium zooplankton filter (MCF), medium zooplankton consumer (MCC), large zooplankton filter (LCF), and large zooplankton consumer (LCC). Among them, there were six FG protozoa, five FG rotifers, and five FG cladocerans and copepods. Fifteen FGs were detected in all seasons. There were 13 to 16 ZFGs detected in Chaohu Lake and the seven other rivers (Table S4). The density and biomass of each ZFG were significantly different. The densities of PA/PB, RF, RSG, PB, and PF, which were the dominant FGs according to their density, were the highest. The dominant ZFGs remained roughly stable across all the seasons. PA/PB (35.01%), RF (22.56%), and RSG (13.85%) were the dominant FGs in spring; PA/PB (24.38%), RF (25.68%), and RSG (26.80%) were the dominant ZFGs in summer; PA/PB (40.99%), RF (20.23%), RSG (8.10%), and PB (11.55%) were the dominant ZFGs in autumn; and PA/PB (43.23%), RF (9.98%), RSG (15.70%), and PF (17.12%) were the dominant ZFGs in winter (Figure 3A). PA/PB was the dominant ZFGs in all seasons. Compared to the other three seasons, the RF density in winter was the lowest. The densities of PB and PF were highest in autumn and winter, respectively (Figure 3A). The densities of PA/PB, RF, and RSG were high in the Chaohu Lake and the seven rivers. The densities of PB and SCF in the Chaohu Lake were high. The densities of PR in the NFR and PR were higher than those in the other rivers (Figure 3C). The densities of RLG and PF were the highest in the HBR. RF, RSG, and protozoa feeding on algae, bacteria, and debris were the main ZFGs in Chaohu Lake and the rivers. The SCF was also the main ZFG in Chaohu Lake.

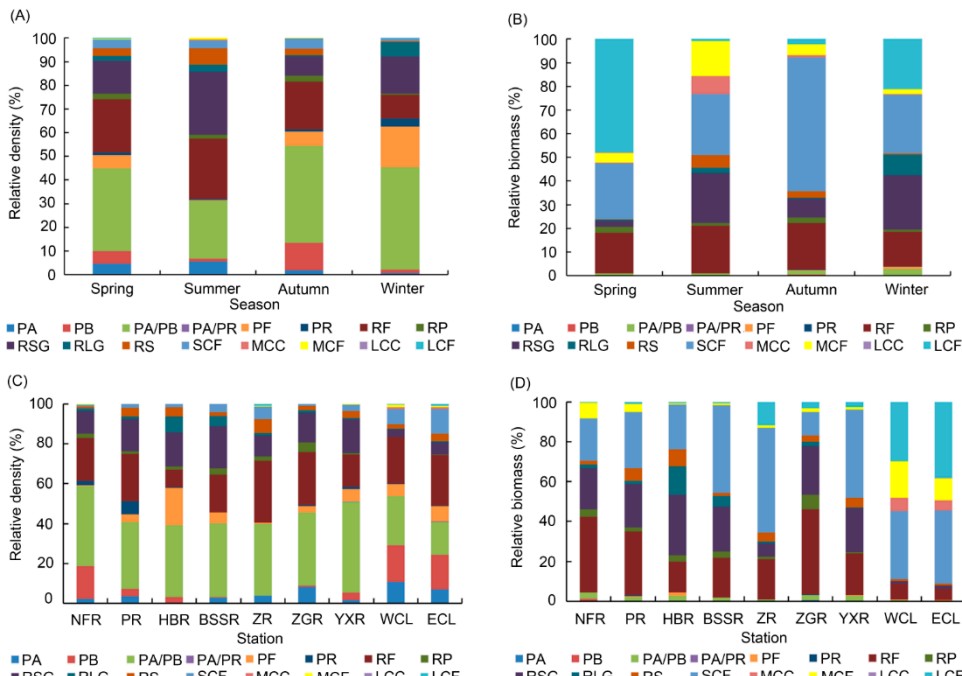

**Figure 3.** Seasonal and spatial variations in density and biomass of zooplankton functional groups (ZFGs). (**A**) seasonal variations in density of ZFGs, (**B**) seasonal variations in biomass of ZFGs, (**C**) spatial variations in density of ZFGs, and (**D**) spatial variations in biomass of ZFGs.

RF, RSG, SCF, MCF, and LCF exhibited the highest biomass and were the dominant ZFGs according to biomass. The dominant ZFGs according to biomass also remained roughly stable across seasons, whereas the relative biomass of these dominant ZFGs had obvious seasonal fluctuations. RF (17.26%), SCF (23.63%), and LCF (47.93%) were the dominant ZFGs according to biomass in spring; RF (20.15%), SCF (25.68%), RSG (21.03%), and MCF (14.86%) were the dominant ZFGs in summer; RF (19.69%), SCF (56.61%), and RSG (7.89%) were the dominant ZFGs in autumn; and RF (14.54%), SCF (25.00%), RSG (22.87%), and LCF (21.11%) were the dominant ZFGs in winter (Figure 3B). The biomasses of LCF and SCF were the highest in spring and autumn, respectively. The biomasses of RF, RSG, and SCF in all seven rivers were high. MCF biomass in the NFR and PR was higher than that in the other rivers. The biomasses of RLG and PF were highest in the HBR. The ZFGs with the highest biomass in the Chaohu Lake were SCF, MCF, and LCF, followed by MCC, and RF. RF, RSG, and SCF were the main ZFGs according to the biomass in the rivers. Filter-feeding cladocerans and copepods of different sizes were the main ZFGs according to the biomass in Chaohu Lake (Figure 3D). The densities and biomasses of filter-feeding cladocerans and copepods in Chaohu Lake were higher than those in the rivers (Figure 3).

Co-occurrence network analysis of species showed that there were significant co-occurrence relationships between ZFGs and RF, with the exception of a significant negative correlation between copepodid and *Bosmina fatalis* in RF, while others were significantly positively correlated (Figure 4A). However, no significant co-occurrence relationship was found between other ZFGs of rotifers and ZFGs of cladocerans and copepods (Figure 4A). It is worth noting that not all species showed co-occurrence relationships according to their ZFGs, which suggested that only the individual size and feeding habits of zooplankton could not fully reflect the ecological relationship of these species (Figure 4A). Co-occurrence network analysis of the sampling sites based on zooplankton species showed that the river sampling sites formed a network. The Chaohu Lake samples in autumn formed a network alone, while the other lake samples in the other three seasons formed a network (Figure 4B). Furthermore, the co-occurrence network built using zooplankton species showed that sampling sites were roughly gathered according to season (Figure 4B). However, the co-occurrence network constructed based on ZFGs showed that sampling sites were not noticeably clustered according to season, especially in summer and autumn (Figure S1). Moreover, the co-occurrence network showed that samples at the same water quality level tended to gather together (Figure 4C). These results imply that using ZFGs as a water quality indicator can effectively eliminate the interference of seasons.

Cluster analysis based on the density of ZFGs showed that the samples collected in spring were divided into four groups. The samples collected from the ZR, BSSR, and ZGR were clustered into a group; those from the WCL, ECL, PR, and YXR were clustered into one group; and the samples from the NFR and HBR were divided into two completely different groups (Figure 5A). The samples collected during summer were divided into four groups. The samples collected from WCL and ECL were clustered into a group; those from the NFR and PR were clustered into a group; the samples collected from the ZR, BSSR, and ZGR were clustered into a group; and the samples collected from the HBR and YXR were clustered into a group (Figure 5B). The samples collected in autumn were divided into three groups. The samples collected from the NFR, WCL, and ECL were clustered into groups. The samples collected from the PR, HBR, and YXR clustered into a group, and the ZR, BSSR, and ZGR clustered into a group (Figure 5C). The samples collected during winter were divided into three groups. The samples collected from the NFR and PR were clustered into one group. The samples collected from the ZR, BSSR, and HBR clustered into one group, and those from the WCL, ECL, ZGR, and YXR clustered into one group (Figure 5D). The ZFGs in Chaohu Lake and its surrounding rivers exhibited clear spatiotemporal heterogeneity. The composition of ZFGs in Chaohu Lake was clearly different from those in the rivers. There were also other obvious differences in the ZFGs between rivers. The ZR, BSSR, and ZGR had similar ZFG compositions in spring, summer, and autumn. The NFR and PR had similar ZFG compositions during summer and winter.

The HBR and YXR had similar ZFG compositions in summer and autumn. However, the clustering results for the water samples did not fully reflect water quality (Figure 5).

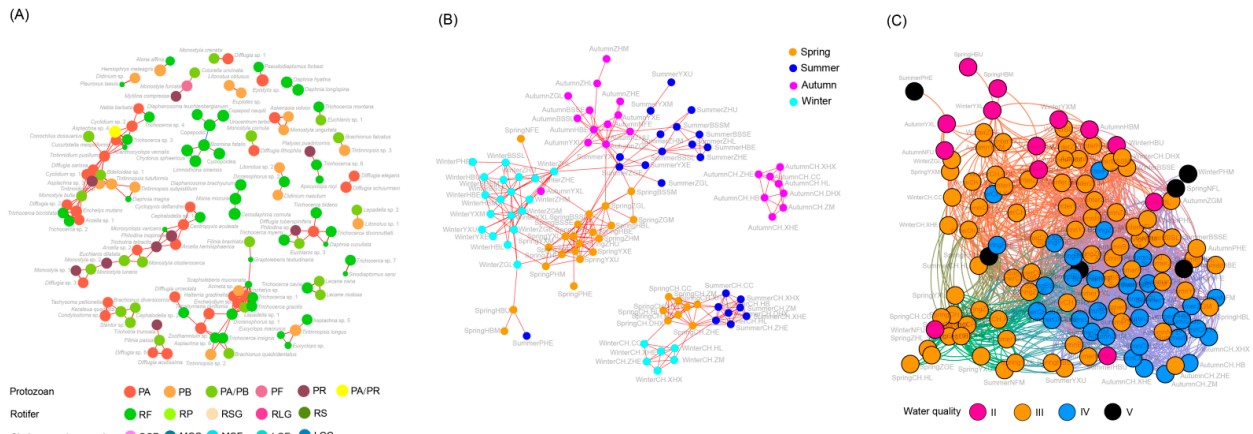

**Figure 4.** Co-occurrence network of zooplankton species (**A**) and sampling sites based on zooplankton species (**B**) and based on zooplankton functional groups (**C**) in Chaohu Lake Basin. Different colors in panel (**A**) indicate different zooplankton functional groups. Different colors in panels (**B**,**C**) indicate different seasons. The red and blue edges indicate significant positive correlation (Spearman correlation coefficient > 0.6 and $p < 0.05$) and significant negative correlation (Spearman correlation coefficient < −0.6 and $p < 0.05$), respectively. The circle diameters in panel (**A**) indicate the relative abundance of the zooplankton species.

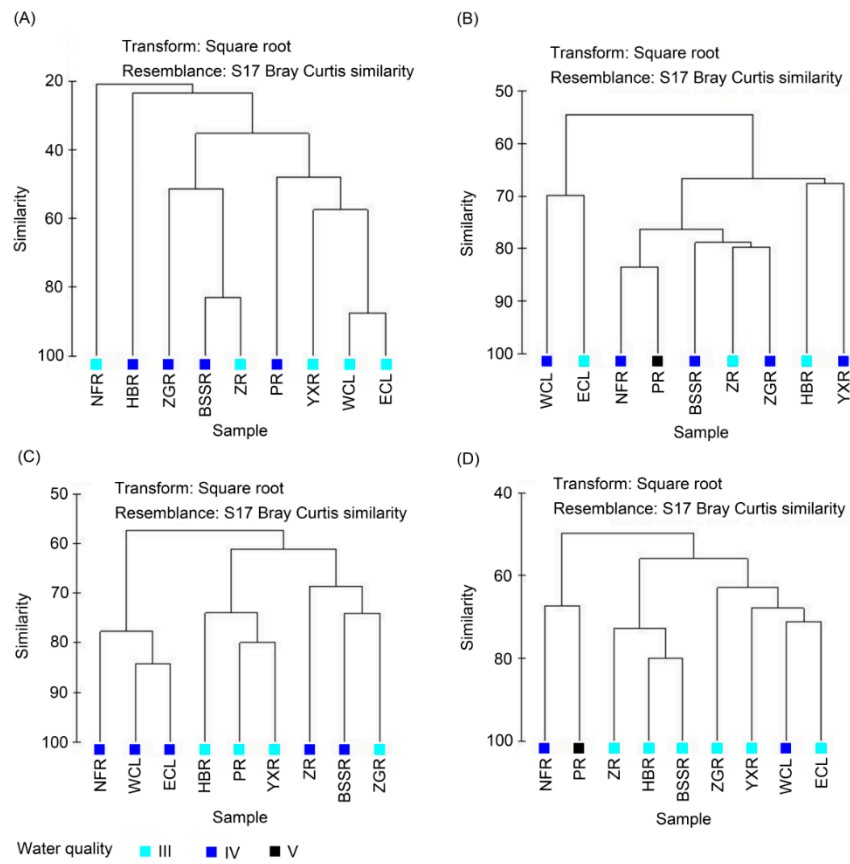

**Figure 5.** Clustering profiles of zooplankton functional groups based on their density. (**A**) spring, (**B**) summer, (**C**) autumn, and (**D**) winter.

### 3.3. Correlation between ZFGs and Environmental Factors

The ecological ZFGs with significant co-occurrence in summer and autumn were significantly higher than those in spring and winter. Using the number of edges of each node as the measurement index, RP, PA/PB, RF, RSG, RLG, LCF, and SCF with three edges were the most important ZFGs in spring. The RSG, RLG, RS, and RF with six edges were the most important ZFGs in summer, followed by MCF and PA/PB with five edges. SCF, MCF, PA, and RF with seven edges were the most important ZFGs in autumn, followed by RSG and RS with six edges and PA/PB and RP with five edges. The SCF with four edges was the most important ZFG, followed by PA/PR, PA, MCC, PF, and RSG with three edges. These results indicate that PA/PB, RF, and RSG played an important role in spring, summer, and autumn for the entire ecosystem. SCF played an important role in spring, autumn, and winter, while the importance of other ZFGs changed with the season. Moreover, the significant co-occurrence patterns of ZFGs changed with season (Figure 6).

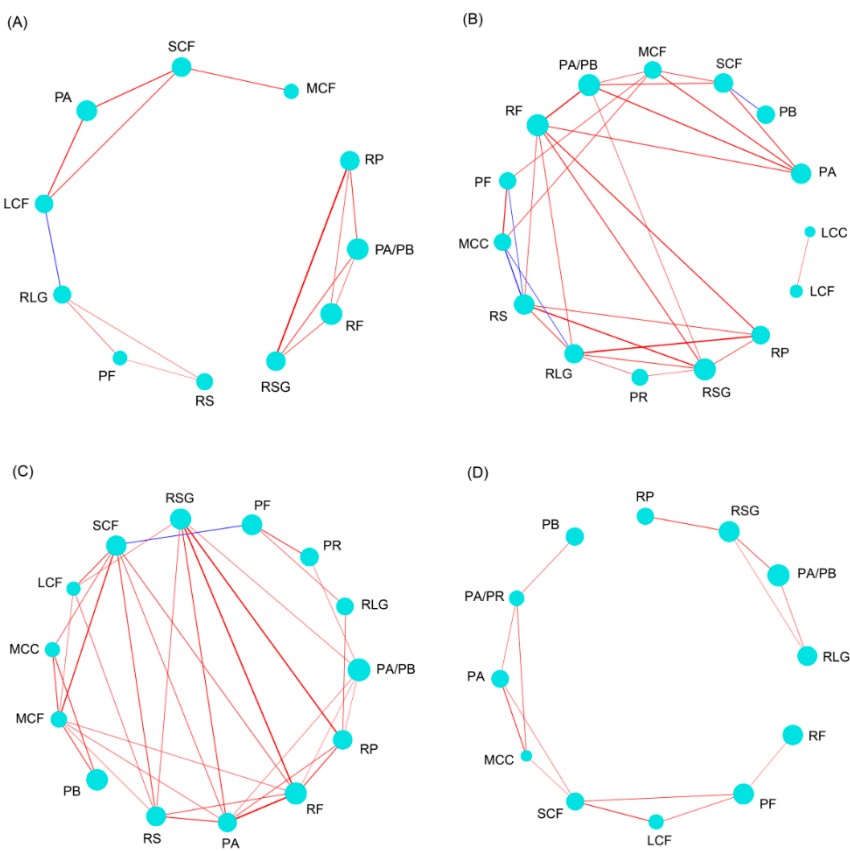

**Figure 6.** Co-occurrence networks of zooplankton functional groups. (**A**) spring, (**B**) summer, (**C**) autumn, and (**D**) winter. The red and blue edges indicate significant positive and negative correlations, respectively. The edge wide indicates the correlation coefficient.

The RDA results showed that the environmental factors significantly correlated with ZFGs and differed between seasons (Figure 7). $COD_{Mn}$, $BOD_5$, TN, $NO_3$-N, $NO_2$-N, $PO_4$-P, and Trans significantly correlated with the ZFGs in spring (Figure 7A). $NO_3$-N, Trans, Cond, pH, TP, TN, $COD_{Mn}$, and Chl a significantly correlated with the ZFGs in summer (Figure 7B). Trans, DO, TP, pH, $COD_{Mn}$, $BOD_5$ and Chl a significantly correlated with the ZFGs in autumn (Figure 7C). WT, $NO_3$-N, $NO_2$-N, $NH_4$-N, TN, $PO_4$-P, TP, $BOD_5$, Chl a, DO, and pH significantly correlated with the ZFGs in winter (Figure 7D).

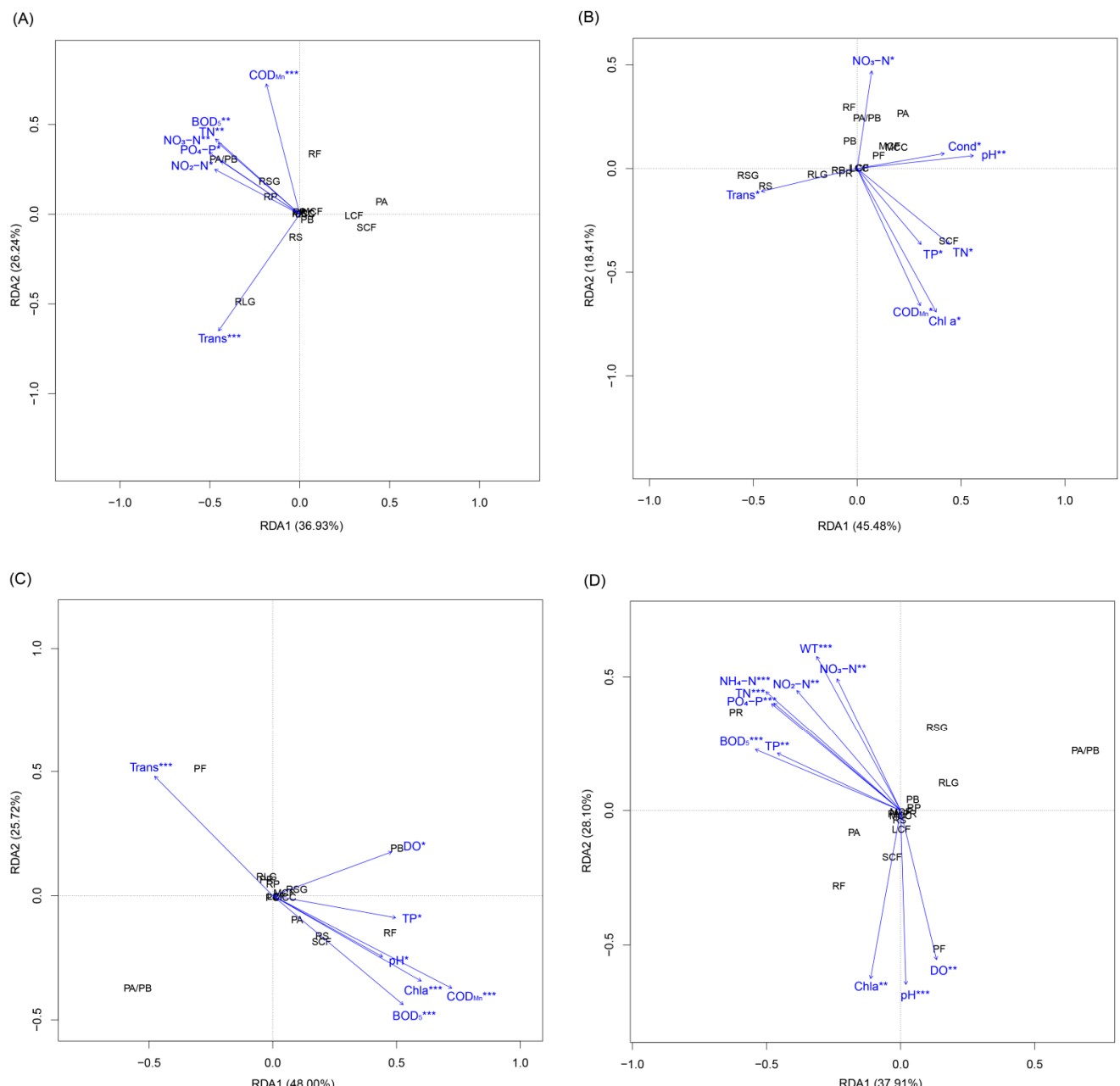

**Figure 7.** Redundancy analysis (RDA) for the zooplankton functional groups and environment factors. (**A**) spring, (**B**) summer, (**C**) autumn, and (**D**) winter. WT, water temperature; DO, dissolved oxygen; Cond, conductivity; Trans, transparency; TN, total nitrogen; TP, total phosphorus; $COD_{Mn}$, permanganate index; $BOD_5$, biochemical oxygen demand after five days; Chl a, chlorophyll-a. * $p < 0.05$; ** $p < 0.01$; *** $p < 0.001$.

Pearson's correlation analysis showed that LCF was significantly negatively correlated with $NO_3$-N. MCC was significantly positively correlated with WT, and significantly negatively correlated with Cond and $BOD_5$. MCF was significantly positively correlated with WT and significantly negatively correlated with pH. PA/PR was significantly positively correlated with DO. PF was significantly positively correlated with Trans, and significantly negatively correlated with WT. PR was significantly positively correlated with $NH_4$-N and $BOD_5$, and significantly negatively correlated with WT. RF and RSG were significantly positively correlated with the WT. RLG was significantly positively correlated with TN and $NO_3$-N and significantly negatively correlated with pH. RS was significantly positively cor-

related with WT and significantly negatively correlated with Cond. SCF was significantly positively correlated with WT, pH, and Chl a and significantly negatively correlated with Cond and NO₃-N (Figure 8 and Table S5).

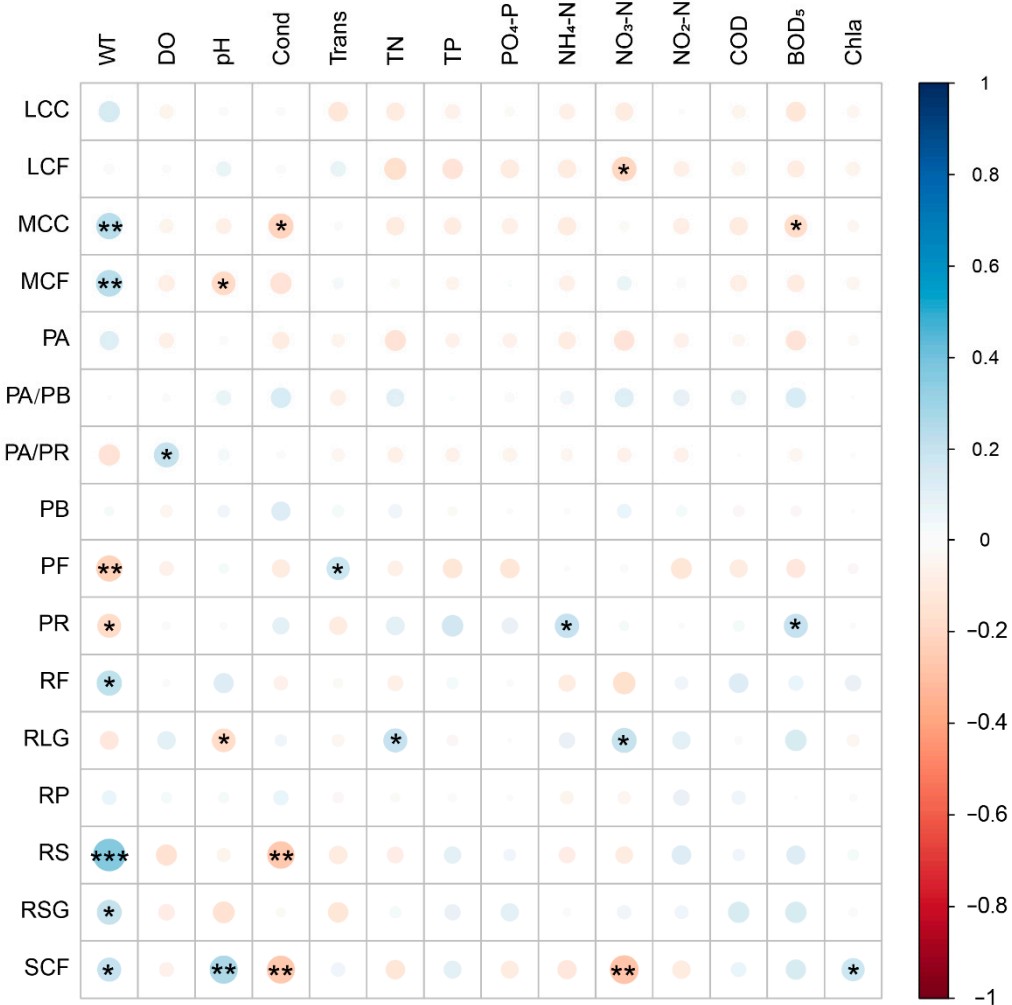

**Figure 8.** Bubble chart shows the Pearson correlation between zooplankton functional groups and environment factors. WT, water temperature; DO, dissolved oxygen; Cond, conductivity; Trans, transparency; TN, total nitrogen; TP, total nitrogen; COD, permanganate index; $BOD_5$, biochemical oxygen demand after five days; Chla, chlorophyll a. * $p < 0.05$; ** $p < 0.01$; *** $p < 0.001$.

## 4. Discussion

Functional traits are the characteristics of species which result from long-term responses and adaptation to their surroundings. Different functional traits reflect different ecological adaptations and species niches. FGs are collections of species with similar ecological functions in a community [31]. Compared with classification methods, FG analysis can help us better understand and predict the relationship between community ecological functions and the environment, as FG research mainly focuses on the ecological functions of species [32,33]. In plankton communities, functional traits mainly include body size, shape, motility, tolerance, and sensitivity to environmental conditions [34]. Currently, dividing aquatic organisms into FGs is the best way to understand their ecological interactions [35]. Body size and feeding style are considered the primary traits and not only include different characteristics related to ecosystem processes but also describe organisms' primary food source; that is, their role in the dynamics of the aquatic food web [35,36]. Obtaining food is one of the basic activities of zooplankton that determine biological adaptability and is

related to a variety of characteristics (including morphological characteristics) [37]. Our results showed that the zooplankton in the Chaohu Lake Basin were divided into 16 ZFGs based on body length and feeding mode. Rotifers and ciliates are considered important predators of bacteria and debris [38,39]. Pratt and Cairns [14] divided nearly 300 protozoa in different habitats, such as wetlands, lakes, rivers, and laboratory microalgae systems, into six ZFGs, of which most species eat bacteria and debris, and reported that most protozoa species participate in the proliferation of bacterial populations and the mineralization of debris carbon in freshwater ecosystems. The prevalence of debris-rich habitats may partly explain the global distribution and ease of maintenance of protozoan species. Finlay and Esteban [40] reported that most species of Ciliophora and Sarcodina feed mainly on bacteria and algae. Rotifers can feed on particles (bacteria and debris) of less than 3 μm in size and algae up to 100 μm in size. The abundance of rotifers and their relative abundance in zooplankton increases with an increase in the abundance of suspended solids [41]. The cladocerans feed primarily through filter feeding by filtering small amounts of food from water, mainly bacteria, and rotten debris. Organic decaying debris is formed by the decomposition of animal and plant residues, as well as animal waste, which is widely distributed in various water bodies. The decaying debris is often accompanied by a large number of bacteria and contains 4.5−5.2 billion bacteria in each gram of wet weight decaying debris in natural freshwater. Therefore, decaying debris is an excellent food source for Cladocera in natural freshwater [25].

Community construction theory holds that environmental selection generally leads to the convergence of functional community traits [31]. Lokko et al. [42] reported significant differences in the functions of rotifer communities in freshwater and brackish water habitats. The differences in the main feeding patterns of rotifers implied that rotifers play different roles in the food web in fresh and brackish water environments. The index based on functional traits can better reveal the pattern change in rotifer community function and the impact of environmental change on the seasonal dynamics of the rotifer community than the index related to morphological classification. ZFGs exhibit clear seasonal changes with changes in environmental conditions [16–18]. In this study, although the number of ZFGs in four seasons and nine sample points in the Chaohu Lake Basin were similar, the ZFGs in the Chaohu Lake Basin had obvious spatiotemporal heterogeneity according to density and biomass. The compositions of the Chaohu ZFGs were different from those of the river ZFGs. There were also obvious differences in the ZFGs between rivers. The community patterns of the ZFGs changed with changes in the environment. In eutrophic freshwater, large zooplankton—such as the Cladocerans filter—feed on algae efficiently. The density and biomass of filter-feeding cladocerans and copepods in the Chaohu Lake were higher than those in the rivers, possibly because the cyanobacterial bloom in the Chaohu Lake provides rich food sources for filter-feeding cladocerans and copepods. Zhang et al. [43] divided the seven rivers in this study into different river types according to the landscape composition of the river basin: the HBR is the forest river type, the NFR and PR are the urban river type, the BSSR and ZR are the agricultural river type, the ZGR is the mixed river type, the YXR estuary and upstream are the mixed river type, and the YXR midstream and downstream are the forest river type. In this study, Chaohu Lake and the seven rivers showed an obvious spatial distribution pattern in the nutrient gradient. The environmental characteristics of Chaohu Lake were different from those of the rivers. Therefore, our results imply that the ZFGs revealed heterogeneity in different environments.

Zooplanktons are important biological monitoring indicators. However, because zooplankton biomass, dominant species, and diversity vary significantly with water conditions such as WT, eutrophication, and pollution, zooplankton species composition cannot accurately reflect water quality [44,45]. Duggan et al. [46] found that the composition of rotifers in water cannot be simply compared with various published rotifer indicator species to determine the nutritional level of water when studying the relationship between rotifer distribution and nutritional level in the northern islands of New Zealand. Lin et al. [47] reported that the species composition of rotifers in various trophic reservoirs is not com-

pletely different. Those considered oligotrophic species are distributed in both mesotrophic and eutrophic reservoirs, whereas those considered eutrophic species are also distributed in oligotrophic reservoirs. Various nutrient indicator species exist in the same reservoir, but their composition proportion differs in different reservoirs. Therefore, to understand the response of aquatic ecosystems to environmental disturbances, it is important to consider functional diversity within the trophic level [48].

Wang et al. [40] reported that water eutrophication can promote the convergence of the functional characteristics of planktonic crustacean communities and that environmental selection is one of the main driving forces of community structure variation. Our results showed that the ZFGs in the Chaohu Lake Basin were affected by changes in environmental factors, and the environmental factors with significant correlation with ZFGs were different across seasons. The distributions of PA/PB, RF, and SCF were closely related to the nutrients $BOD_5$, $COD_{Mn}$ and Chl *a*. Richard et al. [49] reported that TP, TN, DOC, and Chl *a* have significant effects on the functional diversity of crustaceans. Oh et al. [50] reported that water Chl *a*, $COD_{Mn}$, and TP are important environmental factors that affect the functional types of rotifer masticators. Polyarthra density was significantly positively correlated with Chl *a*, $COD_{Mn}$, and TP. Polyarthra abundance increases sharply as water eutrophication rises [41]. Chl *a* represents the biomass of phytoplankton and reflects the primary productivity of the water. $COD_{Mn}$ is a comprehensive index that indicates water organic pollution and prevents water from turning black and smelly [51]. $COD_{Mn}$ indirectly reflects the degree of organic pollution in the water. The $COD_{Mn}$ value is significantly positively correlated with the severity of organic pollution in the water [52]. However, our results showed that $COD_{Mn}$ was not correlated with any ZFGs (Figure 8). $BOD_5$ indirectly reflects the content of biodegradable organic matter in lake water to understand the degree of water pollution. Generally, water with high nutrients, $BOD_5$, $COD_{Mn}$, and Chl *a* has high algae, bacteria, and debris. Therefore, the density of PA/PB, RF, and SCF feeding on algal or bacterial debris suspensions was high. PA/PB, RF, and SCF can be used to characterize the water quality. However, our results showed that PA/PB did not significantly correlate with any environmental factor. RF and SCF were significantly and positively correlated with WT, respectively. Moreover, SCF was also significantly positively correlated with pH and Chl *a* and significantly negatively correlated with Cond and $NO_3$-N. RLG mainly feed on larger food particles, such as flagellates and diatoms. RSG mainly feed on smaller food particles, such as golden algae and cryptoalgae. RF mainly feed on fungi, and RS mainly feed on Trichocerca, which prefers clean water. Therefore, RLG, RS, RSG, and PF can be used to indicate clear water quality. Our results showed that RLG was significantly positively correlated with $NO_3$-N and TN. Only PF was significantly positively correlated with Trans (Figure 8).

Chen et al. [53] reported that WT significantly affected the composition of the rotifer community and its FGs. Obertegger and Flaim [54] reported that the WT is an important environmental predictor of zooplankton taxonomy and functional diversity. Our study showed that although WT had no significant effect on ZFGs in spring, summer, and autumn with a high WT, it had a significant effect on ZFGs in winter with a low WT (Figure 7). However, Pearson's correlation results showed that the number of ZFGs that were significantly related to WT was the largest. PR was not only positively affected by $NH_4$-N and $BOD_5$ but was also negatively affected by WT. The PR density of rivers with high nutrition (NFR and PR) was high, indicating that PR can be used to indicate poor water quality. Compared to spring, summer, and autumn with high WT, winter—with the lowest WT—had the lowest RF density, indicating that RF was not suitable for growth and reproduction in low-temperature water. Oh et al. [50] divided rotifers into eight FGs according to rotifer masticator type and analyzed the effect of rotifers on water quality based on species composition and masticator FGs. Obertegger and Flaim [54] analyzed the effects of environmental factors on the rotifer community according to rotifer feeding and defense FGs. Oh et al. [50] and Wen et al. [55] reported that compared to rotifer species abundance, the relationship between rotifer functional traits and environmental factors

was more obvious. Our co-occurrence network results implied that ZFGs were more stable than zooplankton species across seasons.

## 5. Conclusions

A total of 250 species of zooplankton were identified, including 88 species of protozoa, 115 species of rotifers, 28 species of cladocerans, and 19 species of copepods from Chanhu Lake Basin. Except for the species that could not be accurately divided into ZFGs, 227 zooplankton species were divided into 16 ZFGs. The density and biomass of each ZFG were significantly different. In total, Chaohu Lake and the seven rivers exhibited an obvious spatial distribution in the environmental gradient. The co-occurrence network constructed based on zooplankton species showed that sampling sites roughly gathered according to season. However, the co-occurrence network developed based on ZFGs showed that sampling sites were not clearly clustered according to season, especially in summer and autumn. The ecological ZFGs with significant co-occurrence in summer and autumn were significantly higher than those in spring and winter.

**Supplementary Materials:** The following supporting information can be downloaded at: https://www.mdpi.com/article/10.3390/w14132106/s1, Figure S1: Co-occurrence network constructed based on ZFGs; Table S1: Functional groups of zooplankton; Table S2: Spatiotemporal variations of physicochemical parameters; Table S3: Species compositions of zooplankton detected in this study; Table S4: Compositions of zooplankton functional groups in Chaohu Lake Basin; Table S5. Pearson correlation coefficients between zooplankton functional groups and environment factors.

**Author Contributions:** Conceptualization, L.W. and X.C.; methodology, L.W., L.J., X.C., J.N. and Y.Z.; software, L.W., X.C. and J.N.; validation, Y.Z. and M.G.; formal analysis, L.W. and J.N.; investigation, L.W., L.J., Y.Z. and M.G.; resources, L.W. and X.C.; data curation, Y.Z. and M.G.; writing—original draft preparation, L.W.; writing—review and editing, X.C. and J.N.; visualization, L.W. and J.N.; supervision, Y.Z. and M.G.; project administration, L.W. and X.C.; funding acquisition, L.W. All authors have read and agreed to the published version of the manuscript.

**Funding:** This research was funded by the National Natural Science Foundation of China, grant number 51909051; the Key Laboratory of Ecological Impacts of Hydraulic-projects and Restoration of Aquatic Ecosystem of Ministry of Water Resources; the Project of Ecological Environment Monitoring for Lake Chaohu, grant number CH2022-01; and the Key Natural Science Research Project for Colleges and Universities of Anhui Province, grant number KJ2021A0530.

**Institutional Review Board Statement:** Not applicable.

**Informed Consent Statement:** Not applicable.

**Data Availability Statement:** Physical and chemical factor data can be obtained from the corresponding author. Other data are provided in Supplementary Materials.

**Conflicts of Interest:** J.N. is an employee of Guangdong Meilikang Bio-Science Ltd., China.

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
