# Peer review of "Distribution of Zooplankton Functional Groups in the Chaohu Lake Basin, China"

_water, doi:10.3390/w14132106_

Round 1
Reviewer 1 Report
The paper "Distribution Characteristics of Zooplankton Functional Groups 2 in the Chaohu Lake Basin, China" presented to me for evaluation is of a monitoring nature. It covers the issue of zooplankton abundance against the background of changing physicochemical conditions of water against the background of four seasons. The topic of the work is interesting from the point of view of the correlation of the occurrence and abundance of living organisms depending on environmental conditions. Zooplankton is an important link in the water ecosystem. It is used for monitoring of standing and flowing surface waters. The authors present the results of annual studies in the area of ​​the lake and its catchment area. The study lacked a more detailed description of the area in terms of the presence and displacement of pollutants. It is worth specifying what type of catchment it is - agricultural, industrial or definitely natural. It is also worth listing the potential sources of pollution in the catchment area and trying to make at least a short comparison with the presence and abundance of species against this background. It could be write in introduce. Overall, the work is very valuable and can be further processed after completion.Author Response
Thank you very much for reviewing our manuscript and providing valuable comments. These comments are great value for us to revise the manuscript. We have supplemented the detailed introduction of the existence and discharge of pollutants in Chaohu Lake Basin and the types of catchments in the Introduction section according to your comments.
Reviewer 2 Report
The structural characteristics of zooplankton functional groups and their correlation with environmental physicochemical factors in the Chaohu Lake Basin were analyzed. The main methods used, results obtained, data analysis and implications of the findings are clearly explained.
Suggestions
Tittle, delete Characteristics
Line 39, delete ....the characteristics of
Line 133, What criteria was used for the classification of water types?
Line 178, How did biomass analyze? It is not explained in Methodology
Line 201, Bosmina fatalis in cursiva
Fig. 7, This figure can be replaced by a single figure comparing the different seasons
Fig. 8, to replace by a table
The text in results should be summarized
The discussion should be improved. Some paragraphs are results
Author Response
Comment
Tittle, delete Characteristics
Response
We have deleted the word according to your comment.
Comment
Line 39, delete ....the characteristics of
Response
Thank you for your comment. We have deleted the words according to your comment.
Comment
Line 133, What criteria was used for the classification of water types?
Response
Water quality was classified as per the Environmental Quality Standards for Surface Water (GB 3838-2002), which can be downloaded from the website of the Ministry of Ecology and Environment of the People’s Republic of China (https://english.mee.gov.cn/Resources/standards/water_environment/quality_standard/200710/t20071024_111792.shtml).
Comment
Line 178, How did biomass analyze? It is not explained in Methodology
Response
The biomass was represented by the individual count of each species. The analyzed methods of biomasses of protozoa, rotifer, cladocerans, and copepods were described in the Species Identification and Counting subsection. The biomass of each ZFG was calculated by adding the count of the included species. We have added the description in the Division of ZFGs subsection of our revised manuscript.
Comment
Line 201, Bosmina fatalis in cursiva
Response
We have revised the Bosmina fatalis in italics.
Comment
Fig. 7, This figure can be replaced by a single figure comparing the different seasons
Response
Thank you for your comment. Usually, we put all species and environmental data together for RDA. However, this study focused on the analysis of the relationship between water environmental factors, especially the environmental factors closely related to water quality and zooplankton. Because the impact of water temperature on zooplankton was too strong, if the data collected throughout the whole year were put together, the results would be greatly disturbed by water temperature. We think that the change of water temperature in four seasons should not be used as a key water quality indicator, although the impact of water temperature on zooplankton needs to be considered at the same sampling time. Therefore, we had divided the data into four seasons for conducting RDA.
Comment
Fig. 8, to replace by a table
Response
Thank you very much for your comment. Because the data in the table cannot be placed in the page, we added a supplementary table (Table S5) to show the accurate Pearson correlation coefficients.
Comment
The text in results should be summarized
The discussion should be improved. Some paragraphs are results.
Response
Thank you for your comments. We have revised our manuscript according to your comments.